# JMJ Histone Demethylases Balance H3K27me3 and H3K4me3 Levels at the *HSP21* Locus during Heat Acclimation in *Arabidopsis*

**DOI:** 10.3390/biom11060852

**Published:** 2021-06-07

**Authors:** Nobutoshi Yamaguchi, Toshiro Ito

**Affiliations:** 1Division of Biological Science, Nara Institute of Science and Technology, 8916-5, Takayama, Ikoma-shi, Nara 630-0192, Japan; itot@bs.naist.jp; 2Precursory Research for Embryonic Science and Technology, Japan Science and Technology Agency, 4-1-8, Honcho, Kawaguchi-shi, Saitama 332-0012, Japan

**Keywords:** *Arabidopsis thaliana*, Jumonji, heat acclimation, heat shock protein 21, H3K4me3, H3K27me3

## Abstract

Exposure to moderately high temperature enables plants to acquire thermotolerance to high temperatures that might otherwise be lethal. In *Arabidopsis thaliana*, histone H3 lysine 27 trimethylation (H3K27me3) at the *heat shock protein 17.6C* (*HSP17.6C*) and *HSP22* loci is removed by Jumonji C domain-containing protein (JMJ) histone demethylases, thus allowing the plant to ‘remember’ the heat experience. Other heat memory genes, such as *HSP21,* are downregulated in acclimatized *jmj* quadruple mutants compared to the wild type, but how those genes are regulated remains uncharacterized. Here, we show that histone H3 lysine 4 trimethylation (H3K4me3) at *HSP21* was maintained at high levels for at least three days in response to heat. This heat-dependent H3K4me3 accumulation was compromised in the acclimatized *jmj* quadruple mutant as compared to the acclimatized wild type. JMJ30 directly bound to the *HSP21* locus in response to heat and coordinated H3K27me3 and H3K4me3 levels under standard and fluctuating conditions. Our results suggest that JMJs mediate the balance between H3K27me3 and H3K4me3 at the *HSP21* locus through proper maintenance of H3K27me3 removal during heat acclimation.

## 1. Introduction

High temperature negatively affects plant growth and development [1], and plants possess various defense mechanisms against heat. *Arabidopsis* species have two main heat tolerance mechanisms: basal and acquired heat tolerance [2,3,4]. Basal heat tolerance is the ability to cope with thermal stress when the heat is first sensed. Over several days of heat stress, plants memorize this heat experience and develop the ability to respond to future heat stresses; this is acquired heat tolerance. Thus, exposure to high temperature enables plants to acquire thermotolerance for subsequent higher temperatures.

Acquired heat tolerance requires changes in epigenetic states, which result in differences in expression of heat-resistant genes without changes in the primary DNA sequence. These epigenetic changes include DNA (e.g., cytosine methylation) and histone modifications (e.g., methylation, acetylation, phosphorylation, ubiquitination), which affect the chromatin structure [5,6]. The role of histone methylation and chromatin structure in acquired heat tolerance has been studied. Upon exposure to initial heat, the heat-inducible transactivator heat shock factor A2 (HSFA2) transiently binds to genes encoding heat shock proteins (HSPs) which protect cellular proteins from denaturation [7,8,9]. This transient binding allows target genes to accumulate histone H3 lysine 4 dimethylation (H3K4me2) and trimethylation (H3K4me3) for a few days [9,10,11]. Sustained H3K4me2 and H3K4me3 function as transcriptional heat stress memory [9]. For proper maintenance of stress memory, inactive histone marks and negative histone marks such as histone H3 lysine 27 trimethylation [5] (H3K27me3) might play a role. Such interactions during heat acclimation have not been studied.

Five *Arabidopsis* Jumonji C domain-containing proteins (JMJs), JMJ30, JMJ32, early flowering (ELF), relative of elf (REF6), and JMJ13, are reported to have H3K27 demethylase activity [12,13,14,15,16,17,18]. We recently reported that JMJ30, JMJ32, REF6, and ELF6 are redundantly required for heat acclimation [19]. After exposure to moderate temperature, JMJs affect the expression of several *HSP* genes, including *HSP21* [20], and remove H3K27me3 at the *HSP22* and *HSP17.6C* loci [21]. Sustained H3K27me3 demethylation at those two heat memory genes was maintained for at least three days. Upon exposure to subsequent heat, JMJs directly reactivate *HSP22* and *HSP17.6C* expression through removal of H3K27me3. At the *HSP22* and *HSP17.6* loci, H3K4me3 levels are correlated with *HSP* transcription and anticorrelated with H3K27me3 levels. These results suggest that not only sustained H3K27me3 demethylation, but also sustained H3K4me3 methylation functions as a form of heat memory together with JMJs. However, many JMJ target(s) likely remain to be identified.

## 2. Materials and Methods

### 2.1. Plant Materials

All *Arabidopsis thaliana* lines used in this study were in the Columbia (Col-0) background. The *jmj30-2 jmj32-1 elf6-1 ref6-3* quadruple mutant was described previously [19]. Prior to growth, genotypes were confirmed by PCR using Emerald Amp polymerase (Takara, Japan) (Appendix A). Primers for genotyping were as follows: *jmj30-2* genotyping-FW, CAAACTCTGCTGCAATCGATTTC; *jmj30-2* genotyping-RV, GAAAATGTCACAAGCTCTTGCTTC; *jmj32-1* genotyping-FW, GACTGAGAAAACCTGAACTCAGC; *jmj32-1* genotyping-RV, GTCGTGTAAAGGACTGAAGGTTG; *elf6-1* genotyping-FW, GTCAATGCGGTAATCATTCTAGG; *elf6-1* genotyping-RV, ATATCGAAAATCGAAAAGGAAGC; *ref6-3* genotyping-FW, TCATATACAAGGCGTTCGGTC; and *ref6-3* genotyping-RV, CAGTTGCAACTCTGGAGAAGG. The *jmj30-2* mutant is in the GK line, while *jmj32-1*, *elf6-1*, and *ref6-3* mutants are in the SALK line. The primers to confirm transgenes were as follows: GK-LB-8409, ATATTGACCATCATACTCATTGC; SALK-LBa1, TGGTTCACGTAGTGGGCCATCG.

### 2.2. Normal Growth and Heat Stress Conditions

Procedures for preparation of half-strength Murashige and Skoog (MS) plates and seed surface sterilization were described previously [19]. Prior to heat stress, all the plants were grown on MS plates for four days at 22 °C in a growth chamber under continuous light conditions after stratification at 4 °C for three days. Heat stress treatment was conducted using a water bath shaker (TAITEC, Japan). The plates were placed in resealable zipper storage bags (S.C. Johnson & Son, Racine, WI, USA) and submerged in the water bath. For heat stress memory conditions, four-day-old seedlings were subjected to an acclimation heat of 37 °C for 20 min and then returned to normal growth condition at 22 °C for three days. Subsequently, acclimatized seven-day-old seedlings were subjected to a tester heat stress of 43.5 °C for 20 min and returned to normal growth condition at 22 °C for three days (+ACC+HS).

To grow plants under the conditions that recapitulate those in Nara, Japan (34°41.6′N 135°49.6′E) from 4 August to 10 August 2018, the seeds were sown as described above. The plants were grown for four days after germination in a growth chamber at 22 °C under continuous light. Then, the plants were moved and grown in an SGCmini growth chamber (Clockmics Inc., Tokyo, Japan) under Nara conditions. Past temperature data for Nara were obtained from the Japan Meteorological Agency (https://www.data.jma.go.jp/obd/stats/etrn/index.php (accessed date 5 January 2021)). A map was made by leaflet maps in R.

### 2.3. Estradiol Treatment

Beta-estradiol was dissolved in dimethyl sulfoxide (DMSO) and kept at −30 °C until use. For the mock treatment, the same amount of DMSO without β-estradiol was used as the control. The plants were grown on an MS medium without β-estradiol and then transplanted onto an MS medium with 10 µM β-estradiol using ethanol-sterilized forceps in a hood. Once we started the treatment, the plants were transplanted every two days.

### 2.4. Chromatin Immunoprecipitation (ChIP)

Procedures for the ChIP assay were described previously [19]. We fixed 100–300 mg tissues by formaldehyde for 15 min. After quenching, the resulting tissues were frozen in liquid nitrogen and kept at −80 °C until use. The tissues were homogenized in a nuclei extraction buffer and the chromatin protein solution was obtained after removal of debris by filtering through Miracloth (Merck, Kenilworth, NJ, USA). After fragmentation by an Ultrasonic Disruptors UD-201 sonicator (Tomy, Tateishi, Japan), immunoprecipitation was conducted using H3K4me3 (ab8580; Abcam, UK), H3K27me3 (ab6002; Cambridge, UK), H3 antibodies (ab1791; Abcam, UK), or HA (12CA5; Roche, Basel, Switzerland) and Dynabeads Protein A or G (Thermo Fisher Scientific, Waltham, MA, USA). The resulting beads were washed two times each with a low-salt buffer, a high-salt buffer, a LiCl buffer, and a TE buffer. Chromatin was eluted from beads overnight at 65 °C. The resulting chromatin was purified using a QIAquick PCR Purification Kit (Qiagen, Venlo, The Netherland). DNA was quantified with a LightCycler 480 (Roche, Switzerland) using the FastStart Essential DNA Green Master mix (Roche, Switzerland). The ratio of ChIP over input DNA (% input) was compared based on the reaction threshold cycle for each ChIP sample compared to a dilution series of the corresponding input sample. Three independent experiments were performed. Statistical significance was computed using either the one-way ANOVA test followed by the post-hoc Tukey’s HSD test or the two-tailed Student’s *t*-test for multiple- and single-pair comparisons, respectively.

## 3. Results

### 3.1. Acclimation-Dependent Sustained H3K4me3 Methylation at the HSP21 Locus

To identify target gene(s) with higher H3K4me3 three days after exposure to initial heat, we computationally reanalyzed genome-wide H3K4me3 and RNA sequencing (RNA-seq) data in the wild type and the *jmj30-2 jmj32-1 ref6-3 elf6-1* quadruple mutant (hereinafter referred to as the ‘*jmjq* mutant’) seedlings with and without acclimation [19] (Figure 1a). We previously screened genes that showed no difference in expression just after HS in acclimatized wild type and *jmjq* mutant seedlings but did show significant differences at 4 or 24 h after HS [19]. These analyses identified 80 downregulated genes in the *jmjq* mutant as compared to the wild type [19] (false discovery rate < 0.05) (Figure 1b).

We next identified genes that have acclimation-dependent sustained methylation of H3K4me3 via JMJ proteins. Under the control condition, 119 genes were hypertrimethylated in the wild type as compared to the *jmjq* mutant (Figure 1b). After acclimation, 83 genes had higher H3K4me3 levels in the wild type than in the *jmjq* mutant (Figure 1b). *HSP21* is the only gene that showed increased H3K4me3 levels at three days (72 h) after heat acclimation in the wild type as compared to the *jmjq* mutant. The identification of only one gene could be due to the period after acclimation and/or efficiency of acclimation; H3K4me3 may be mainly used as a heat memory for shorter time periods and/or different temperatures. Nevertheless, we successfully identified *HSP21* as a candidate gene that has JMJ-dependent sustained H3K4me3 methylation three days after acclimation.

ChIP sequencing (ChIP-seq) peaks at the *HSP21* locus showed no significant difference in H3K4me3 levels between the wild type and *jmjq* mutants under control conditions (Figure 1c). Although H3K4me3 at the *HSP21* gene accumulated to high levels in the wild type three days after acclimation, such enrichment was not observed in acclimatized *jmjq* mutants (Figure 1c). H3K27me3 levels remained low in the wild type in both control and acclimation conditions and were higher in *jmjq* mutants regardless of acclimation (Figure 1c). No difference in histone H3 signals was observed between the wild type and *jmjq* mutants with and without acclimation (Figure 1c). These data imply that histone modifications rather than the locations of histones (or nucleosomes) along the DNA are changed due to acclimation.

### 3.2. Proper Maintenance of the Balance between H3K27me3 and H3K4me3 by JMJs

To further understand histone modification dynamics during heat acclimation, we next carried out a time course ChIP–quantitative PCR (qPCR) analysis with acclimation and heat shock (Figure 2a). Based on a previous publication, *HSP21* expression levels increased rapidly upon acclimation and gradually after heat shock in the wild type. In *jmjq* mutants, *HSP21* upregulation was compromised after acclimation and heat shock [19]. Consistent with the transcriptional changes, active H3K4me3 marks in the wild type increased rapidly just after acclimation and remained at high levels for three days (Figure 2b). After heat shock, H3K4me3 levels increased again in the wild type (Figure 2b). H3K4me3 levels in the *jmjq* mutants stayed overall lower than those in the wild type (*p* < 0.05 by Student’s *t*-test) (Figure 2b). H3K27me3 remained at low levels in the wild type at all time points, while the levels in *jmjq* mutants were high (*p* < 0.05 by Student’s *t*-test) (Figure 2c). Histone H3 signals were unchanged in the wild type and *jmjq* mutants (Figure 2d). Considering that JMJs remove H3K27me3, these results suggest that higher H3K27me3 accumulation in *jmjq* mutants inhibits proper accumulation of H3K4me3 at the *HSP21* locus.

To understand whether JMJ30 is directly recruited to the *HSP21* locus, we performed ChIP–qPCR using the *pJMJ30:JMJ30-HA* line [18,19]. In vivo association of the JMJ30 protein with the *HSP21* locus was observed only after acclimation and heat shock (*p* < 0.05 by Student’s *t*-test) (Figure 2e). This binding may contribute to maintaining H3K4me3 levels in response to heat via removal of H3K27me3. Since H3K27me3 levels were higher in the wild type before acclimation than in *jmjq* mutants, JMJ32, REF6, and/or ELF6 might bind to the *HSP21* locus without heat, unlike JMJ30.

### 3.3. Manipulation of Balance between H3K27me3 and H3K4me3 by JMJ30 Induction

Because JMJ30 bound to the *HSP21* locus in response to heat, we hypothesized that JMJ30 induction changes both H3K4me3 and H3K27me3 levels. Our previous study showed that *HSP21* expression was induced in *jmjq* mutants if JMJ30 was induced prior to acclimation [19]. Thus, we reasoned that JMJ30 induction prior to acclimation should remove H3K27me3 and promote accumulation of H3K4me3. Indeed, H3K4me3 in *jmjq* mutants was restored to the wild type levels when JMJ30 was induced prior to acclimation but not before heat shock (*p* < 0.05 by one-way ANOVA test; *p* < 0.05 by the post-hoc Tukey’s HSD test) (Figure 3a,b). Furthermore, induction of JMJ30 prior to acclimation led to decreased H3K27me3 levels in *jmjq* mutants (*p* < 0.05 by one-way ANOVA test; *p* < 0.05 by the post-hoc Tukey’s HSD test) (Figure 3c). Finally, histone H3 levels were unchanged under any of the conditions tested. The data are consistent with the hypothesis that JMJ30 induction prior to acclimation is sufficient to change both H3K4me3 and H3K27me3 levels at the *HSP21* locus in the *jmjq* mutant background.

### 3.4. Maintenance of Balance between H3K27me3 and H3K4me3 by JMJs under Field Conditions

Previous studies suggested that JMJs remove H3K27me3 at the *HSP22* and *HSP17.6C* loci not only under laboratory conditions, but also under fluctuating field conditions [19]. We hypothesized that JMJ activities are required for acclimation-dependent H3K4me3 association at the *HSP21* locus under fluctuating field conditions as well. To examine this hypothesis, ChIP–qPCR was conducted using wild type and *jmjq* mutant seedlings grown under conditions that replicate those observed in Nara, Japan [19] (Figure 4a).

Before the temperature increase (at 10 am on 9 August 2017), no significant difference between wild type and *jmjq* mutant seedlings was observed in terms of H3K4me3 levels at the *HSP21* locus. After the temperature increase (at 1 pm on 9 August 2017), H3K4me3 rapidly accumulated at the *HSP21* locus in the wild type and exceeded the levels seen in *jmjq* mutants (*p* < 0.05 by one-way ANOVA test; *p* < 0.05 by the post-hoc Tukey’s HSD test) (Figure 4b). Furthermore, H3K27me3 levels were higher in *jmjq* mutants at all the time points as they were under laboratory conditions (*p* < 0.05 by one-way ANOVA test; *p* < 0.05 by the post-hoc Tukey’s HSD test) (Figure 4c). Finally, histone H3 levels were unchanged under all the conditions tested (Figure 4d). These data indicate that JMJ activities are required for acclimation-dependent H3K4me3 association at the *HSP21* locus under fluctuating field conditions.

## 4. Discussion

JMJs directly reactivate the *HSP22* and *HSP17.6C* expression through the removal of H3K27me3 during heat acclimation [19]. Although we had previously revealed that several *HSP* genes other than *HSP22* and *HSP17.6C* were downregulated in *jmjq* mutants, how those genes were regulated in the context of chromatin remained unclear. The results of this study suggest that JMJs mediate the balance between H3K27me3 and H3K4me3 at the *HSP21* locus through proper maintenance of H3K27me3 removal during heat acclimation. Five pieces of evidence support this. First, the expression of *HSP21* was significantly lower in *jmjq* mutants than in the wild type in response to heat [19]. Second, the level of H3K27me3 was significantly higher in *jmjq* mutants than in the wild type due to the failure to remove H3K27me3 in the mutants (Figure 1 and Figure 2). Third, the level of H3K4me3 was significantly lower in *jmjq* mutants than in the wild type (Figure 1 and Figure 2). Fourth, the level of histone H3 was unchanged between the wild type and *jmjq* mutants (Figure 1 and Figure 2). Fifth, JMJ30 directly bound to the *HSP21* locus in response to heat. Therefore, JMJs contribute to maintaining the balance between H3K27me3 and H3K4me3 at the *HSP21* locus.

### 4.1. Transcriptional and Epigenetic Regulation of the HSP21 Gene by H3K27me3 Demethylases during Heat Acclimation

The chloroplast-localized small heat shock protein HSP21 prevents aggregation of thermosensitive client proteins [20,22]. The flexible N-terminal arms of the HSP21 protein interact with client proteins while its C-terminal tails maintain the dodecamer and chaperone activity [22]. Loss and reduction of HSP21 activity decrease survival rate, whereas constitutive overexpression of HSP21 causes thermotolerance [20,23]. HSP21 rapidly accumulates after heat stress and remains abundant even in the absence of heat [20]. HSP21 is involved in plastid-encoded RNA polymerase (PEP)-dependent transcription [23]. HSP21 physically interacts with pTAC5 and controls chloroplast development under heat stress by maintaining PEP function [24,25]. On progression of the memory phase, FtsH6 protease negatively controls HSP21 accumulation [20,26,27,28,29]. Thus, the protein structure, regulation, and function of HSP21 are relatively well-known.

*HSP21* transcription is mainly controlled by transcription factors, such as HSFA1 and HSFA2, and chromatin regulators, such as forgetter (FGT), the SWI/SNF chromatin remodeler brahma (BRM), and the ISWI chromatin remodelers chromatin-remodeling protein 11 (CHR11) and CHR17. *HSP21* expression levels are downregulated in *hsfa2* [8,9], *hsfa1a/b* [30], *fgt*, *brm*, and *chr11 chr17* [31] mutants during heat acclimation. Among those factors, HSFA2 mediates deposition of H3K4me3 upon acclimation (Figure 5). Here, we identified JMJs as regulators of *HSP21* expression during heat acclimation (Figure 5). JMJ30 directly bound to the *HSP21* locus in response to heat. While JMJ30 association at *HSP21* was observed only after heat stress, *jmjq* mutants had excess H3K27me3 at *HSP21* without heat stress (Figure 2). Thus, JMJ30, JMJ32, ELF6, and REF6 are redundantly required to remove H3K27me3 at *HSP21* without heat stress. Alternatively, JMJs antagonistically functions to prevent Polycomb group proteins from depositing H3K27me3 at the *HSP21* locus [5] (Figure 5). H3K27 hypertrimethylation in *jmjq* mutants blocks deposition of H3K4me3. At least one of the JMJ proteins might bind to the *HSP21* locus without heat stress. During plant development, transcription factors act as docking points for the different histone modification enzymes [32]. Such a mechanism might lead to the removal of H3K27me3 and the deposition of H3K4me3.

Among the four JMJ proteins, JMJ30 directly bound to the *HSP21* locus only in response to heat. One reason why JMJ30 association is enhanced upon heat stress could be stabilization of the *JMJ30* mRNA and the JMJ30 protein [18]. Although *HSP21* expression is strongly upregulated (10,000-fold) [19], stabilization of the *JMJ30* mRNA and the JMJ30 protein is moderate (10-fold). Considering fold changes, the stabilization may not be the only reason for upregulation of the *HSP21* gene. Since many histone modification enzymes interact with transcription factors [33], JMJ30 might also interact with transcription factors for target recruitment. Because transcriptional levels of *HSFA1* or *HSFA2* are upregulated by heat, HSFAs could be interaction partners for JMJ recruitment [7,8,9,30]. Other chromatin regulators might also contribute to JMJ recruitment [31]. Identification of interaction partners of JMJs will be needed to better understand the molecular mechanism of JMJ30 recruitment in response to heat.

### 4.2. Transcriptional and Epigenetic Regulation of the HSP21 Gene under Natural Conditions

Previous studies suggested that the differences in gene expression and histone modification between the wild type and the *jmjq* mutants were larger under fluctuating field temperature conditions than they were under laboratory conditions [19]. Like *HSP22* and *HSP17.6C*, *HSP21* expression is regulated by JMJ functions in response to uniformly high temperature and naturally fluctuating temperatures [19]. H3K27me3 levels were higher in *jmjq* mutants at all the time points as they were under laboratory conditions. We also observed subtle but significant differences in H3K27me3 levels in *jmjq* mutants between 10 am and 1 pm. After 3-h exposure to heat, H3K27me3 levels were downregulated in *jmjq* mutants at 1 pm compared to the levels at 10 am. This result suggests that additional, redundant H3K27me3 demethylase(s) still function in the *jmjq* mutant background. JMJ13 is one candidate since it also acts as a temperature-dependent H3K27me3 demethylase [15,34,35]. H3K27me3 removal might allow deposition of H3K4me3 as seen in *jmjq* mutants at 1 pm. Overall, JMJs mediate a balance between H3K27me3 and H3K4me3 at the *HSP21* locus for heat adaptation.

Field-grown *Arabidopsis halleri* plants express *Ahg945273* (the *HSP21* ortholog) in summer but not in winter [36,37]. Thus, *HSP21* expression could be regulated by a conserved mechanism in *Arabidopsis thaliana* and *Arabidopsis halleri*. *Flowering locus C* (*FLC*) expression is regulated by shared epigenetic mechanisms between those two plant species [38,39,40]. Comparing the regulatory logic between *Arabidopsis thaliana* and different plant species might help to identify unique and shared heat adaptation mechanisms in plants.

### 4.3. The Redundant JMJ Network Regulating HSP Genes

A previous study showed differences in phenotypic severity between *jmjq* mutants and higher-order *hsp17.6c hsp22* double mutants [19]. Acclimation defects in *jmjq* mutants were stronger than in *hsp17.6c hsp22* mutants. This result suggested that JMJ downstream genes other than *HSP17.6C* and *HSP22* must exist for heat acclimation. In this study, we found that *HSP21* could be such a target. It will be worthwhile to create *hsp17.6c hsp22 hsp21* triple mutants to confirm the role of *HSP21* during heat acclimation. Furthermore, the rescue of *jmjq* mutants by overexpressing HSP21 could reveal the importance of *HSP21* upregulation in response to heat. JMJs mediate a balance between H3K27me3 and H3K4me3 at the *HSP21* locus through proper maintenance of H3K27me3 removal during heat acclimation. In *Arabidopsis*, *Arabidopsis trithorax* (ATX) proteins play key roles [41,42,43]. Elucidating the molecular mechanisms by which H3K4me3 levels at *HSP21* are controlled by ATXs remains an exciting challenge for the future.

## Figures and Tables

**Figure 1 biomolecules-11-00852-f001:**
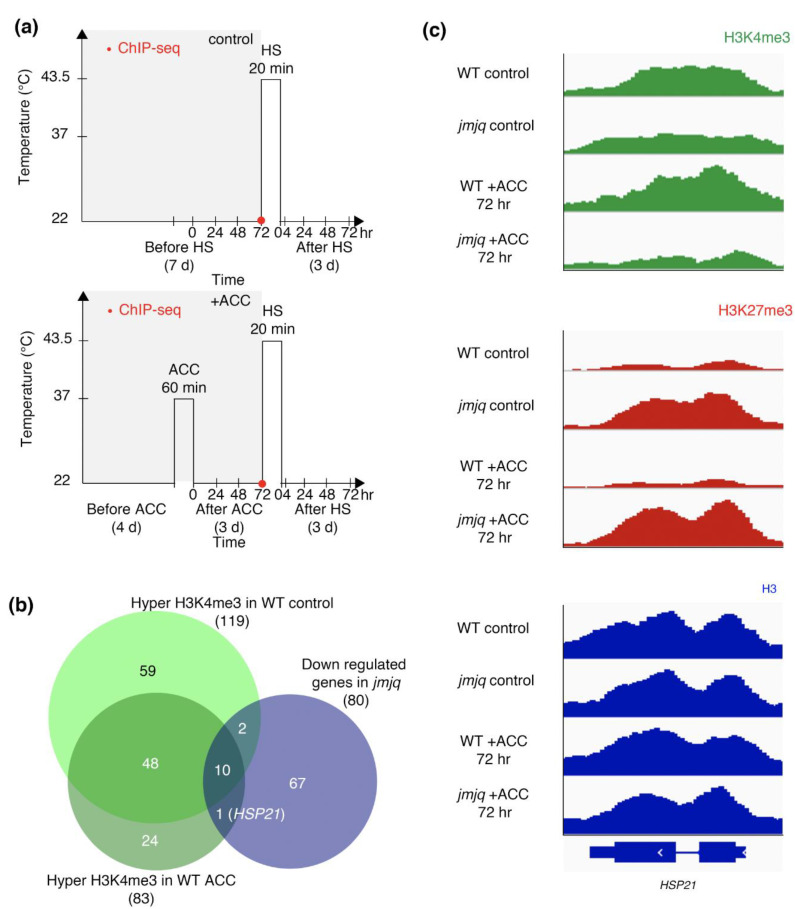
Jumonji demethylases are required for proper H3K4me3 accumulation in response to acclimation (ACC). (**a**) Schematic representation of the temperature conditions used. Timing of ChIP-seq is denoted by red dots. Above, basal thermotolerance condition; below, heat stress memory condition. (**b**) Venn diagrams showing the overlap between the genes downregulated in *jmjq* mutants and the genes with elevated H3K4me3 in the wild type (WT) with and without acclimation compared to the equivalent *jmjq* mutants (*p* = 2.3 × 10^−17^ for elevated H3K4me3). (**c**) H3K4me3, H3K27me3, and H3 peaks at the regulatory regions of the *HSP21* gene in the wild type and *jmjq* mutants without and with acclimation.

**Figure 2 biomolecules-11-00852-f002:**
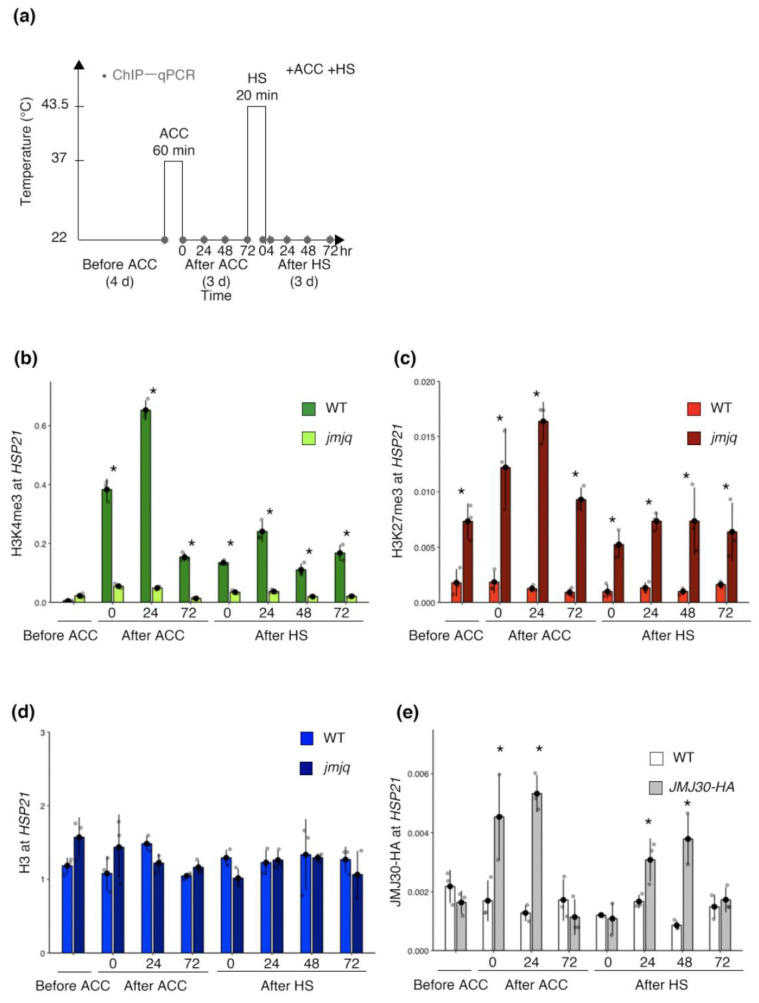
Jumonji 30 directly bound to the *HSP21* locus in response to heat for maintenance of the H3K4me3 and H3K27me3 balance. (**a**) Schematic representation of the temperature conditions used. Timing of ChIP–qPCR is denoted by grey dots. (**b–e**) H3K4me3 (**b**), H3K27me3 (**c**), and histone H3 (**d**) levels in the wild type (WT) and *jmjq* mutants, and JMJ30-HA (**e**) levels at the *HSP21* gene as determined by ChIP–qPCR. Gray jitter dots represent the expression level in each sample. Asterisks indicate significant difference at 0.05 levels based on Student’s *t*-test between the wild type and *jmjq* mutants at the same time point. NS, not significant; ACC, acclimation; HS, heat shock.

**Figure 3 biomolecules-11-00852-f003:**
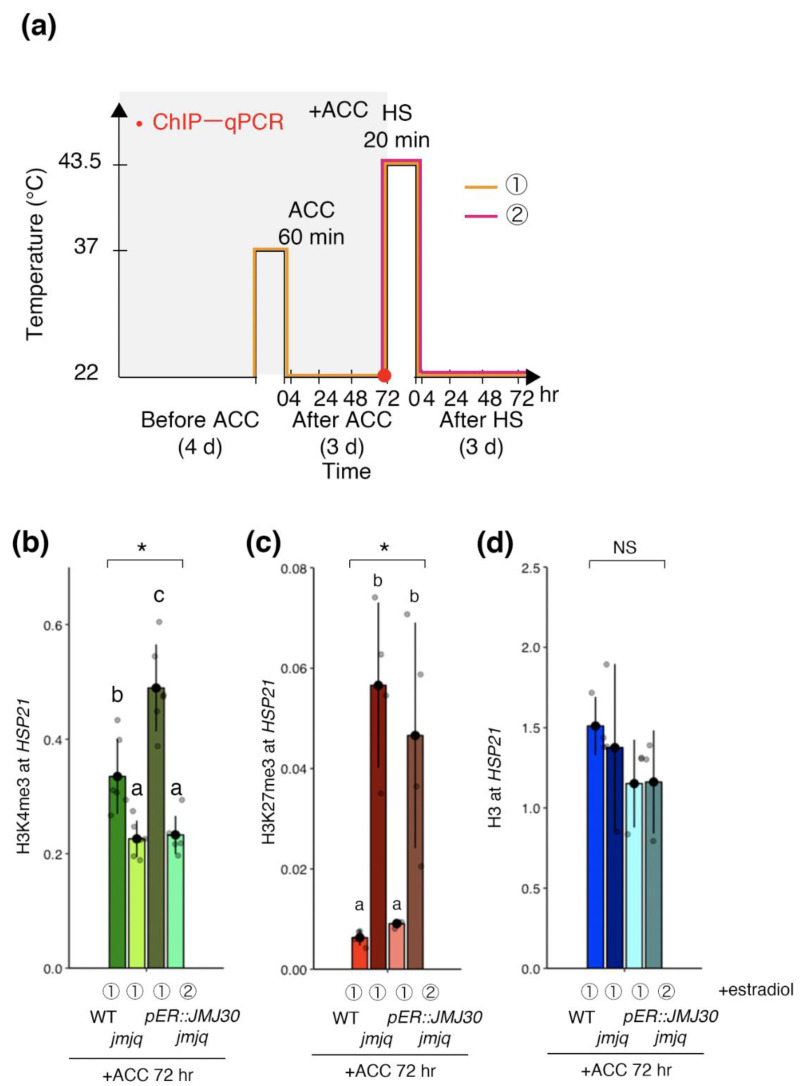
Jumonji 30 demethylase is sufficient for the removal of histone H3K27me3 from the *HSP21* gene before heat shock and affects H3K4me3 levels. (**a**) Schematic representation of the temperature conditions used. Timing of ChIP–qPCR is denoted by the red dot. Orange and magenta lines show two different timings of β-estradiol application. Transgenic plants (*pER8:JMJ30*) in the *jmjq* mutant background subjected to β-estradiol application before acclimation (ACC) (①, left) and before heat shock (HS) (②, right). (**b**–**d**) H3K4me3 (**b**), H3K27me3 (**c**), and histone H3 (**d**) levels at the *HSP21* gene in the wild type and *jmjq* mutants as determined by ChIP–qPCR. Gray jitter dots represent expression level in each sample. Asterisks indicate significant differences based on one-way ANOVA test (* *p* < 0.05). The letters above the bars indicate significant differences, while the same letters indicate non-significant differences (post-hoc Tukey’s HSD test (*p* < 0.05)). NS, not significant.

**Figure 4 biomolecules-11-00852-f004:**
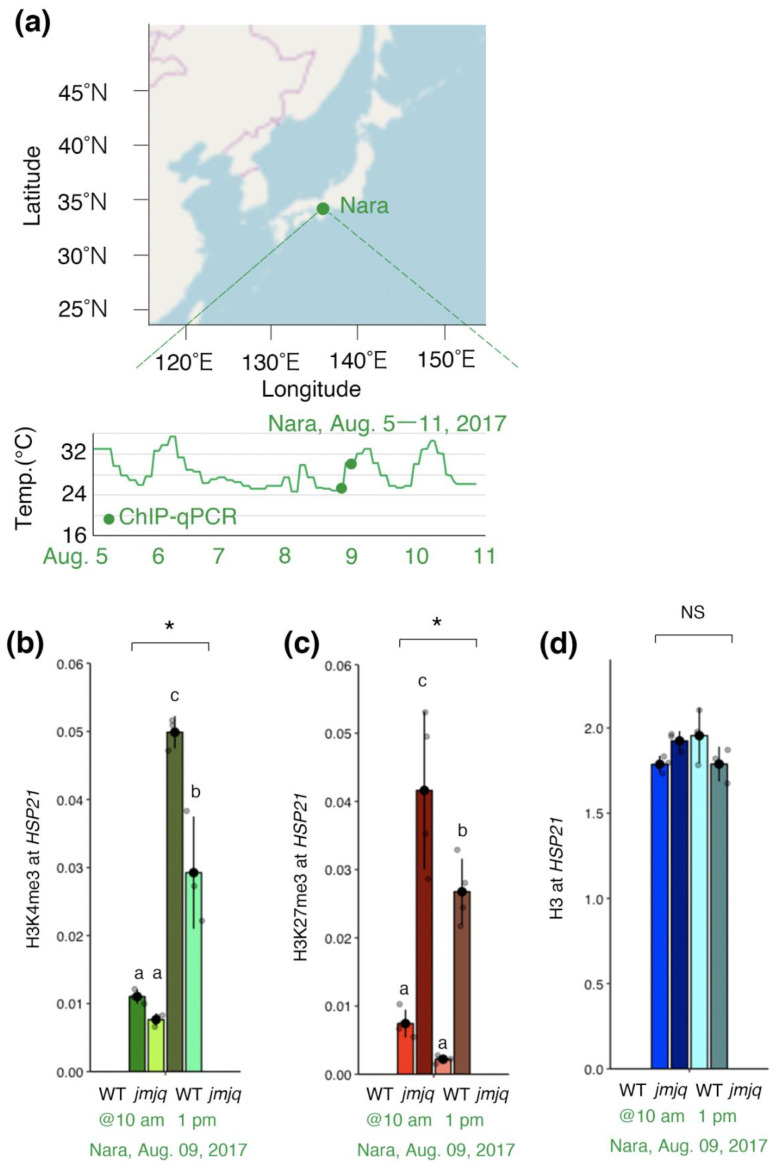
Jumonji demethylases are required for the removal of histone H3K27me3 from the *HSP21* gene under the fluctuating Nara conditions. (**a**) Location of the city of Nara in Japan (above). Schematic representation of the temperature conditions in Nara (below). Timing of ChIP–qPCR is denoted by the green dot. (**b**–**d**) H3K4me3 (**b**), H3K27me3 (**c**), and histone H3 (**d**) levels at the *HSP21* gene in the wild type and *jmjq* mutants as determined by ChIP–qPCR. Gray jitter dots represent the expression level in each sample. Asterisks indicate significant differences based on one-way ANOVA test (* *p* < 0.05). The letters above the bars indicate significant differences, while the same letters indicate non-significant differences (post-hoc Tukey’s HSD test (*p* < 0.05)). NS, not significant.

**Figure 5 biomolecules-11-00852-f005:**
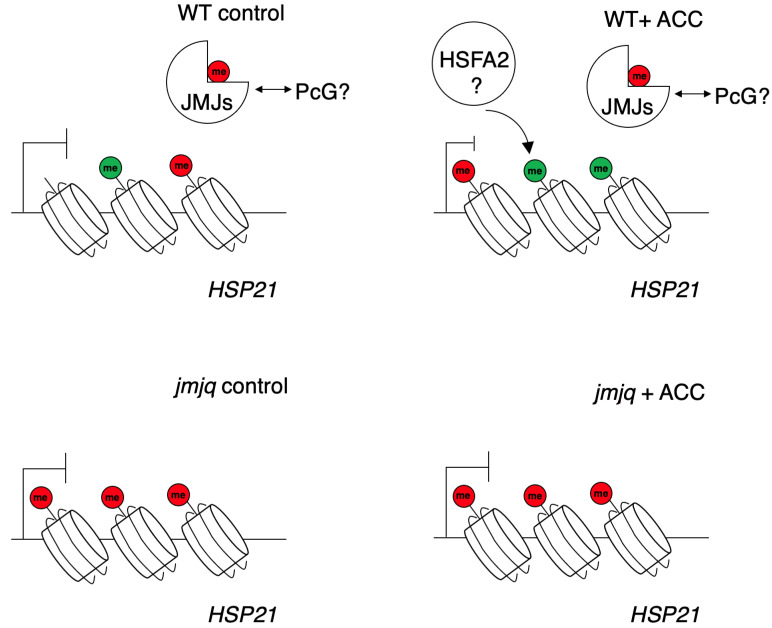
The current model of JMJ-mediated balance between H3K27me3 and H3K4me3 at the *HSP21* locus. Three red and green circles represent H3K27me3 and H3K4me3, respectively. JMJs are required to sustain H3K4me3 methylation. In *jmjq* mutants, higher H3K27me3 accumulation inhibits proper accumulation of H3K4me3 at the *HSP21* locus.

## Data Availability

Not applicable.

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
