# Peer review of "JMJ Histone Demethylases Balance H3K27me3 and H3K4me3 Levels at the HSP21 Locus during Heat Acclimation in Arabidopsis"

_biomolecules, 2021, doi:10.3390/biom11060852_

Round 1

Reviewer 1 Report

In their manuscript, Yamaguchi and Ito continue previous work (Yamaguchi et al., 2019) to reveal the mode plants memorize heat experience. Exposure to moderate temperature enables plants to acquire thermotolerance for subsequent lethal high temperature. In Arabidopsis, histone H3 lysine 27 trimethylation (H3K27me3) at the HEAT SHOCK PROTEIN 17.6C (HSP17.6C) and HSP22 loci are removed by histone demethylases JUMONJI-C DOMAIN-CONTAINING PROTEINs (JMJs) to memorize heat experience. Although other heat memory genes, such as the HSP21 gene are downregulated in acclimatized jmj quidruple mutants compared to wild type, it is not clear how these genes are regulated. Herein the authors showed  that histone H3 lysine 4 trimehylation (H3K4me3) at HSP21 is kept at higher levels for at least three days in response to heat. But, this heat-dependent H3K4me3 accumulation was compromised in acclimatized jmjq mutant compared to acclimatized wild type. JMJ30 directly binds to the HSP21 locus in response to heat and coordinates H3K27me3 and H3K4me3 levels under both lab and fluctuating conditions. The authors suggest that JMJs mediate a balance between H3K27me3 and H3K4me3 at the HSP21 locus through proper maintenance of H3K27me3 removal during heat acclimation.

The work is of interest. It is a follow-up to recently published work from same lab. Probably the amount of new information is not sufficient for a full  publication. They could   have submitted it to be published as a Short Communication. At any case, the work is well planned and performed and the manuscript well presented.

Minor points

Page 3/line 136, please write ..due to the..

Page 3/line 140, please write ..showed no...

Page 5/line 180 and actually throughout the text, the authors instead of..bind(s).., write ..bound... Please correct.

Reviewer 2 Report

Manuscript title:

JUMONJI-mediated balance between H3K27me3 and H3K4me3 at HEAT SHOCK PROTEIN 21 locus during heat acclimation.

General comments:

Authors have characterized that JUMONJI-C DOMAIN-CONTAINING PROTEINs (JMJs) mediate a balance between H3K27me3 and H3K4me3 at the HSP21 locus through proper maintenance of H3K27me3 removal during heat acclimation. Especially, JMJ30 directly bound to the HSP21 locus in response to heat and coordinates H3K27me3 and H3K4me3 levels under both lab and fluctuating conditions. Also, their previous study showed that JMJs directly reactivate HSP22 and HSP17.6C expression through removal of H3K27me3 during heat acclimation.

Authors have clearly mentioned about the materials and methods and drawn the conclusions.

Major comments:

  1. Based on the results of the study, a schematic or pictorial description as a Figure should be added that clearly explains what the authors claim. Especially, it should be well expressed that JMJs-mediated balance between H3K27me3 and H3K4me3 at HEAT SHOCK PROTEIN 21 locus during heat acclimation plays an important role in the plant heat stress response.

Minor comments:

  1. Needs English correction.
  2. Overall, resolutions of figure images should be enhanced. For examples, font sizes of titles, values in graphs should be enlarged and adjusted.
  3. Give attention on typing errors.
  4. Line no 15.- “quadruple” ?. Use the correct word.
  5. Line no 67.- “The jmj30-2 jmj32-1 elf6-1 ref6-3 quadruple mutant was described previously.” For this, please provide references.
  6. Line no 68.- “Prior to growth, genotypes were confirmed by PCR using Emerald Amp polymerase”. The real data should be provided as Supplementary information.
  7. Line no 105.- “quenching” ?. Use the correct word.
